# Sort and Sieve: Pre-Triage Screening of Patients with Suspected COVID-19 in the Emergency Department

**DOI:** 10.3390/ijerph18179271

**Published:** 2021-09-02

**Authors:** Kirsten R.C. Hensgens, Inge H.T. van Rensen, Anita W. Lekx, Frits H.M. van Osch, Lieve H.H. Knarren, Caroline E. Wyers, Joop P. van den Bergh, Dennis G. Barten

**Affiliations:** 1VieCuri Medical Centre, Department of Emergency Medicine, 5912 BL Venlo, The Netherlands; ivrensen@viecuri.nl (I.H.T.v.R.); alekx@viecuri.nl (A.W.L.); dbarten@viecuri.nl (D.G.B.); 2VieCuri Medical Centre, Department of Intensive Care, 5912 BL Venlo, The Netherlands; 3VieCuri Medical Centre, Department of Clinical Epidemiology, 5912 BL Venlo, The Netherlands; fvosch@viecuri.nl; 4School of Nutrition and Metabolism (NUTRIM), Maastricht University, 6229 ER Maastricht, The Netherlands; cwijers@viecuri.nl (C.E.W.); jvdbergh@viecuri.nl (J.P.v.d.B.); 5VieCuri Medical Centre, Department of Internal Medicine, T5912 BL Venlo, The Netherlands; lknarren@viecuri.nl

**Keywords:** emergency department, COVID-19, pandemic, pre-triage screening, triage

## Abstract

Introduction. To reduce the risk of nosocomial transmission, suspected COVID-19 patients entering the Emergency Department (ED) were assigned to a high-risk (ED) or low-risk (acute medical unit, AMU) area based on symptoms, travel and contact history. The objective of this study was to evaluate the performance of our pre-triage screening method and to analyse the characteristics of initially undetected COVID-19 patients. Methods. This was a retrospective, observational, single centre study. Patients ≥ 18 years visiting the AMU-ED between 17 March and 17 April 2020 were included. Primary outcome was the (correct) number of COVID-19 patients assigned to the AMU or ED. Results. In total, 1287 patients visited the AMU-ED: 525 (40.8%) AMU, 762 (59.2%) ED. Within the ED group, 304 (64.3%) of 473 tested patients were COVID-19 positive, compared to 13 (46.4%) of 28 tested patients in the AMU group. Our pre-triage screening accuracy was 63.7%. Of the 13 COVID-19 patients who were initially assigned to the AMU, all patients were ≥65 years of age and the majority presented with gastro-intestinal or non-specific symptoms. Conclusion. Older COVID-19 patients presenting with non-specific symptoms were more likely to remain undetected. ED screening protocols should therefore also include non-specific symptoms, particularly in older patients.

## 1. Introduction

### Background

The COVID-19 pandemic, caused by severe acute respiratory syndrome coronavirus 2 (SARS-CoV-2), has placed an ongoing burden on healthcare systems globally [1,2]. During the first COVID-19 wave in the Netherlands, which lasted from 27 February 2020 (first confirmed infection) to 30 June 2020, there were 50,262 confirmed cases of infection (of which 11,877 were hospitalised) and 6113 confirmed COVID-19-deaths [3].

As the most common access point for acute hospital care, emergency departments (EDs) have experienced several challenges since the outbreak began in Wuhan, China in December 2019 [4,5,6,7]. EDs are not only confronted with a surge of severely ill patients, they also play an indispensable role in the early identification and subsequent segregation of COVID-19 suspected patients [8,9]. This screening process is crucial to lower the risk of nosocomial transmission and to guard the safety of patients and staff [10,11].

However, a lack of knowledge on clinical presentation impeded the identification and segregation of COVID-19 patients. Early in the pandemic, EDs were forced to work with case definitions that focused on acute respiratory illness and travel or contact history [2,12,13]. With increasing exposure to this new infectious disease, it became apparent that COVID-19 patients could be relatively asymptomatic in the early stages of illness or present with gastro-intestinal or atypical symptoms [14,15,16,17,18]. Ultimately, screening methods were highly susceptible to these changes.

Various approaches to the screening and segregation of ED patients have been reported. Many hospitals included the set-up of external facilities with pre-triage prior to ED entrance in order to isolate high-risk patients in an early stage [19,20,21,22]. Others adjusted existing in-hospital areas into “high- and low-risk” zones, with screening mainly based on the presence or absence of fever and/or respiratory complaints [6,23,24].

To date, validated research on the performance of these approaches is limited, which complicates incorporation of these strategies by other EDs. Most previous studies focused on positive versus negative results of patients with suspected COVID-19 infection [21,23]. However, the patients that are initially screened negative (not COVID-19 suspected and therefore not isolated), but later appear to have a (asymptomatic or presymptomatic) COVID-19 infection may pose the greatest risk of nosocomial spread. With this study we aimed to provide more insight into the COVID-19 patients that are likely to remain undetected upon ED entrance, thereby ultimately optimising ED screening.

In our hospital, patients were assigned on arrival (pre-triage) to a high-risk or low-risk area in the ED. Assignment was based on latest guidelines concerning symptoms, known COVID-19 contacts and travel history. The objective of this study was to evaluate the performance of this pre-triage screening method for suspected COVID-19 patients. Additionally, we analysed the characteristics of patients who were assigned to the low-risk area but later received a COVID-19 diagnosis that could be linked to the Acute Medical Unit-Emergency Department (AMU-ED) visit.

## 2. Materials and Methods

### 2.1. Study Design and Setting

We performed a retrospective, observational study in a Dutch teaching hospital. The study period was from 17 March 2020 until 17 April 2020. Our hospital was situated in a high-prevalence region for COVID-19, with COVID-19 admission rates ranging from 90–330 per 100,000 inhabitants [3].

Anticipating an increased demand for emergency care in our hospital, it was decided to deploy the acute medical unit (AMU), an observational unit for patients during the first 24 h of admission, as an extension of the ED. The conjoined AMU-ED was segregated into a high-risk and a low-risk area: both medically unstable (for adequate resuscitation cubicles) and COVID-19 suspected patients were assigned to the ED (high-risk area) and isolated, whereas low-acuity, COVID-19 non-suspected patients were assigned to the AMU (low-risk area). This specific approach was comprehensively described in a previous publication [25].

Pre-triage screening was applied for possible COVID-19 infection based on the most recent guidelines concerning symptoms, known COVID-19 contacts and travel history [13]. A patient suspected for COVID-19 could be identified by the referring general practitioner (GP), ambulance personnel or an ED nurse. The attending emergency physician supervised the final decision on the patient’s allocation area. Figure 1 shows our pre-triage screening patient flowchart.

The COVID-19 case definition in the Netherlands was determined by the Dutch National Institute for Public Health and Environment [13]. Early in the study period, the case definition was relatively strict, including only patients with fever, respiratory complaints and a positive contact and/or travel history. Later in the study period, the case definition was further extended based on latest evidence, also including patients with gastro-intestinal and non-specific symptoms [26]. Non-specific symptoms included loss of smell and taste and muscle ache.

Personal protective equipment (PPE) was used in both low- and high-risk areas according to the latest national guidelines, which were frequently adapted throughout the course of the study period [27]. Furthermore, it was recommended to adhere to basic hygienic measures and physical distancing policy as much as possible.

The testing policy included reverse transcription polymerase chain reaction (RT-PCR) assays of nasopharyngeal swab specimens of COVID-19 suspected patients. Due to nationwide testing capacity shortages, it was not feasible to test all patients (whether COVID-19 suspected or not) attending the conjoined AMU-ED. The decision to perform a RT-PCR test in the ED or AMU was based on the physician’s clinical suspicion (according to the latest national guidelines) and the need for hospital admission. RT-PCR tests were taken upon admission. At that time, it took approximately 24 h before results became available, which therefore did not affect the decision-making process in the AMU-ED.

Patients that were suspected of COVID-19 but had no need for hospital admission were sent home with quarantine advice according to the national guidelines [28].

This study was approved by the institutional review board of VieCuri Medical Centre. Due to the retrospective and observational approach of the study, a waiver of informed consent was provided. This study is reported in accordance with the Strengthening the Reporting of Observational studies in Epidemiology (STROBE) guidelines [29].

### 2.2. Patients

All patients ≥ 18 years presenting to the AMU-ED between 17 March and 17 April 2020 were included for analysis. The screening process for eligibility was conducted by two independent researchers (K.R.C.H. and I.H.T.R.). In case of doubt, a third researcher (D.G.B.) was consulted.

### 2.3. Data Collection and Outcome Measures

Data were collected by two independent researchers (K.R.C.H. and I.H.T.R.) from electronic medical patient records and included demographical data, triage level (Manchester Triage System (MTS); the MTS allocates patients to five urgency categories, which determine the maximum time to the first contact with a physician: red is immediate, orange is very urgent, yellow is urgent, green is standard and blue is non urgent) [30], primary specialty, symptoms, vital signs in the ED, SARS-CoV-2 RT-PCR assay of nasopharyngreal swab specimens, admission rates, in-hospital mortality within 30 days, development of possible COVID-19 symptoms with additional RT-PCR assays during the hospital admission and hospital personnel absenteeism registries. Data were anonymously stored in SPSS, version 25 (IBM Corp., Armonk, NY, USA).

The primary outcome was the number and proportion of patients with a COVID-19 diagnosis in the AMU or ED. A COVID-19 diagnosis was defined as a positive result on a RT-PCR assay of nasopharyngeal swab specimens. Additionally, we analysed characteristics of patients that were initially assigned to the low-risk area but later received a COVID-19 diagnosis that could be linked to their AMU-ED visit.

### 2.4. Statistical Analysis

Descriptive statistics are presented as means with standard deviation (SD), medians with interquartile ranges (IQRs) or percentages, and differences in patient characteristics between AMU and ED were tested using Mann–Whitney’s U or chi-square tests. Pre-triage screening performance (expressed as sensitivity, specificity, positive predictive value (PPV), negative predictive value (NPV) and accuracy) was assessed by means of a 2 × 2 table comparing the presence of COVID-19 based on a positive PCR with the patient’s allocation area (ED vs. AMU). The results were presented with 95% confidence intervals. A *p*-value < 0.05 was considered statistically significant. Statistical analyses were performed using IBM SPSS Statistics, version 25 (IBM Corp., Armonk, NY, USA) and MedCalc.

## 3. Results

Between 17 March and 17 April 2020, 1287 patients visited the AMU-ED. Of these, 525 (40.8%) patients were assigned to the AMU and 762 (59.2%) to the ED. Baseline characteristics are summarised in Table 1.

Compared to patients assigned to the AMU, those in the ED were older (69 vs. 60 years, *p* < 0.001) and more frequently male (58.1% vs. 50.7%, *p* = 0.008). Patients in the ED presented significantly more often with respiratory symptoms and non-specific symptoms, whereas patients assessed in the AMU more often had symptoms related to trauma, the gastro-intestinal tract or the neurological tract. Patients allocated to the ED were more frequently triaged as urgent (red/orange) compared to patients in the AMU, who were two-fold more often triaged as non-urgent (green/blue). Admission and in-hospital mortality rates were significantly higher for ED patients compared to patients in the AMU.

Among the 762 patients assigned to the ED, 473 (62.1%) had a RT-PCR test performed, which was positive in 304 (64.3%) patients. Twenty-eight of 525 AMU patients (5.3%) had a RT-PCR test performed, of which 13 (46.4%) were found to be COVID-19 infected.

The 28 AMU patients were RT-PCR tested because of the following reasons: respiratory symptoms (*n* = 8), gastro-intestinal symptoms (*n* = 4), non-specific symptoms (*n* = 4), infiltrative anomalies on chest X-ray or computed tomography (CT) (*n* = 5), clinical deterioration (*n* = 1), AMU revisit within 24 h (*n* = 3), pre-operative screening (*n* = 2) and transfer to another hospital (*n* = 1).

Our pre-triage screening performance resulted in a sensitivity of 95.9% (93.1–97.8%), a specificity of 8.2% (4.6–13.1%) and an accuracy (overall probability that a patient is correctly classified) of 63.7% (59.3–67.9%) (Table 2).

Of the 191 admitted AMU patients, 28 patients had a RT-PCR test performed during the AMU visit, as stated previously. Of the remaining 163 patients, 151 (92.6%) developed no additional symptoms during the hospital stay that could indicate a pre- or asymptomatic presentation during the AMU visit. Of the 12 patients (7.4%) that developed possible COVID-19 symptoms during admission, nine had a RT-PCR test performed. Six patients tested negative and three patients tested positive for a COVID-19 infection. Only one of these three patients (0.6% of the 163 admitted AMU patients) developed symptoms within the 14 day incubation period, starting from the day of the AMU visit. The remaining three patients who developed fever or respiratory symptoms, had no RT-PCR test undertaken due to another probable diagnosis other than COVID-19. Based on hospital personnel outbreak registries, we did not experience any COVID-19 infection that could be linked to a mutual spread among patients and personnel within the ED-AMU department.

Table 3 shows the baseline characteristics of patients that were initially allocated to the AMU, but later received a COVID-19 diagnosis that could be linked to the index visit. Of these 13 patients, the median age was 76 years and 53.8% were male. Ten patients (76.9%) presented during the first 2 weeks of the study period. Seven patients (53.8%) presented with gastro-intestinal symptoms and six (46.2%) with non-specific symptoms.

## 4. Discussion and Limitations

### 4.1. Discussion

EDs are a major entry point for acute hospital care and therefore have a crucial role in the identification and subsequent segregation of COVID-19 suspected patients. In this study, we assessed the performance of our pre-triage screening method based on symptoms, known COVID-19 contacts and travel history during the first wave of the pandemic. Of all patients allocated to the ED (high-risk area), 62% had a RT-PCR test performed, of which 65% were positive. Of the 525 patients assigned to the AMU (low-risk area) 5% had a RT-PCR test performed, of which 46% were found to be positive. Of the latter, all patients were 65 years or older and most patients presented with gastro-intestinal or non-specific complaints.

As expected with the high COVID-19 prevalence in this region, we found high COVID-19 rates (64.3%) in our ED patients. These findings are in line with studies from Italy (50%) [31] and Belgium (32.7%) [20], two European countries with a similar COVID-19 prevalence during the first wave [32,33]. By contrast, several other studies reported lower COVID-19 rates in ED patients, ranging from 3.8% to 13.7% [21,23,34,35]. In addition to epidemiological and demographic variations, these numbers may be explained by differences in hospital referral thresholds, lockdown regimes and ED utilisation rates for non-urgent complaints. Furthermore, triage and testing policies may have differed between EDs.

Patients assigned to our ED were generally older and predominantly male compared to patients in the AMU. These findings are consistent with demographical data from China [16,36]. In addition, the high- and low-risk groups in our study showed major differences in presenting complaints. As expected, patients in the ED presented most often with respiratory symptoms (71.3%), whereas patients in the AMU mostly presented with symptoms due to trauma (40.8%) or related to the gastro-intestinal tract (21.1%). An Italian study with a comparable design (Turcato et al.) showed similar results [31]. Their differences in triage urgency were also similar to our findings. In contrast, Fistera et al. observed that COVID-19 patients were more often triaged as non-urgent compared to non-COVID-19 patients [21]. However, this study only assessed patients with symptoms suggestive of COVID-19, which complicates comparison with our cohort.

Because of limited evidence on the performance of screening methods in previous studies, we aimed to determine the accuracy of our pre-triage screening protocol. Ideally, the lowest number of possible COVID-19 infected patients were allocated to the low-risk area (AMU). With a high sensitivity (95.9%) and a low specificity (8.2%), our pre-triage screening method reflected a cautious approach. This resulted in “unnecessary” isolation in 30% of ED patients with a negative RT-PCR. However, this strategy based on an overly strict degree of suspicion has to be preferred during the age of a novel infectious disease that dictates a low threshold for strict isolation. Consistent with a low specificity, the overall accuracy did not exceed 63.7%. Of course, the accuracy calculation was based on the COVID-19 positive RT-PCR tests in both groups, rather than the actual COVID-19 prevalence. If more AMU patients had tested negative, the accuracy and performance of our screening protocol most likely would have been higher.

Two Italian studies, one focusing on physician’s gestalt in suspected COVID-19 patients and one similar pre-triage screening method, found higher diagnostic performance numbers with a sensitivity of 83–91% and a specificity of 79–95% [31,37]. The results of Turcato et al., however, were calculated based on the total number of patients in both areas rather than the actual numbers of patients that were tested for COVID-19. When recalculated using our method, results slightly differed. Although the specificity increased to 97.9%, the sensitivity declined to 52.6%, with an accuracy of 83.2%. Another contributing factor that may explain differences in screening protocol performance is the lower ED referral threshold in Italy compared to that of the Dutch health care system, which is well known for the strong gatekeeper role of general practitioners. Consequently, the ED in the Italian study may have dealt with fewer patients who were likely to have a COVID-19 infection.

Thirteen patients (2.5% of all AMU patients) that were initially assigned to the low-risk area were later found to be COVID-19 positive. This pre-triage failure rate shows that our method was not perfect. However, it should be acknowledged that this pre-triage screening protocol was introduced in an early phase of the pandemic, when there was little knowledge about non-specific features of the disease. During the course of the study period, screening protocols were gradually adapted to the latest available evidence. This is also reflected by the fact that 10 of these 13 COVID-19 patients presented during the first two weeks of the study period. Although protocol accuracy clearly improved by adding non-specific features to the case definition, patients were mainly tested upon clinical suspicion and if patients were admitted. Estimation of protocol accuracy would be most reliable if all patients in the ED and AMU were tested, even in non- and low-suspect cases, but due to test capacity shortages this was not feasible during this phase of the pandemic.

Considering the high COVID-19 prevalence in the ED, it is likely that the true percentage of COVID-19 positive patients in the AMU was higher than we observed due to asymptomatic and presymptomatic COVID-19 patients (studies show a prevalence of between 44% and 62% of the presenting patients [14,38,39]). Data from He et al. showed that the viral load of SARS-CoV-2 was highest on day 0 of symptom onset [40]. Although (pre-)triage screening protocols cannot rely on this substantial transmission potential, it does emphasise the need for valid rapid PCR testing at the beginning of a pandemic in order to improve screening processes.

Although it was not possible to identify asymptomatic patients, we did screen for presymptomatic patients by means of additional follow-up data. We identified only one COVID-19 patient that developed symptoms during the hospital stay that could potentially be linked to a presymptomatic stage during the AMU visit. Moreover, we did not experience any nosocomial COVID-19 infection that could be linked to a mutual spread among patients and personnel within the ED-AMU department. This may be explained by basic hygiene measures, physical distancing and PPE usage, which likely reduced in-hospital spread of COVID-19 [41].

All 13 under-triaged patients were 65 year or older and the majority presented with gastro-intestinal or non-specific symptoms. Previous studies have illustrated that up to 50% of SARS-CoV-2 infections can manifest with gastro-intestinal or atypical symptoms [17,18,42] and that these patients arouse the least suspicion for a COVID-19 infection [43]. In a Belgian screening protocol study, atypical symptoms including confusion, repeated falls and altered general state in patients >75 years of age were additionally considered as signs of possible COVID-19 infection [20]. Their screening protocol for the need of hospitalisation resulted in a sensitivity of 88% and a specificity of 93%. ED screening protocols for COVID-19 should therefore include gastro-intestinal complaints and non-specific complaints, such as falls, altered mental state and malaise, especially in older patients.

### 4.2. Limitations

This study has several limitations. First, because of test capacity shortages we did not routinely perform RT-PCR in both the AMU and ED, so definite diagnosis using the golden standard was lacking in, respectively, 95% and 60% of the patients. However, we evaluated follow-up data of the additional 163 AMU patients that had no RT-PCR test performed in the AMU. During their hospital stay, only one additional patient (0.6%) was diagnosed with a COVID-19 infection. Consequently, 99.4% developed no COVID-19 symptoms or had a negative test result on a RT-PCR nasopharyngeal swab specimen. Furthermore, our study design represents daily practice in many hospitals around the world. During a pandemic, scarce diagnostic tests should be used expediently and one may question the added value of testing asymptomatic or presymptomatic patients in an ED setting [38]. Basic hygienic measures, proper PPE and physical distancing may have more impact [41,44]. Second, this was a single centre study and the applicability of our results to other hospitals may vary. Screening protocol outcomes are affected by a priori chances of the disease of interest, probably resulting in different outcomes in EDs with dissimilar epidemiological situations. Third, this study had a retrospective design with potential limitations in accuracy of reporting. Fourth, we used RT-PCR assays of nasopharyngeal swab specimens as our diagnostic implement, which have a single test sensitivity ranging from 60% to 78% [45]. To achieve maximal sensitivity, some studies therefore suggest to combine a chest CT-scan with PCR sampling [45]. At the time of conducting this study, these recommendations were not yet implemented in our hospital protocol, explaining why we chose to solely base our results on PCR assays. Finally, there is no available perfect approach to managing pandemics. At the beginning of 2020, PPE shortages, hospital bed volumes and limited access to diagnostic tests determined the ED handling of this pandemic. Protocols changed over time and could differ between health systems. For example, some countries developed systems of massive testing, which probably would have changed the outcomes of this study. 

## 5. Conclusions

EDs are a major entry point for acute hospital care and therefore have a crucial role in the recognition and subsequent segregation of patients with possible COVID-19. In the early phase of this new pandemic, with strict case definitions due to a lack of knowledge and limited testing possibilities, our pre-triage screening method showed an accuracy of 64%. Older COVID-19 patients were more likely to present with atypical symptoms and were therefore at risk to remain undetected in the ED. To improve early identification and segregation, ED screening protocols should therefore include non-specific complaints, particularly in older patients.

## Figures and Tables

**Figure 1 ijerph-18-09271-f001:**
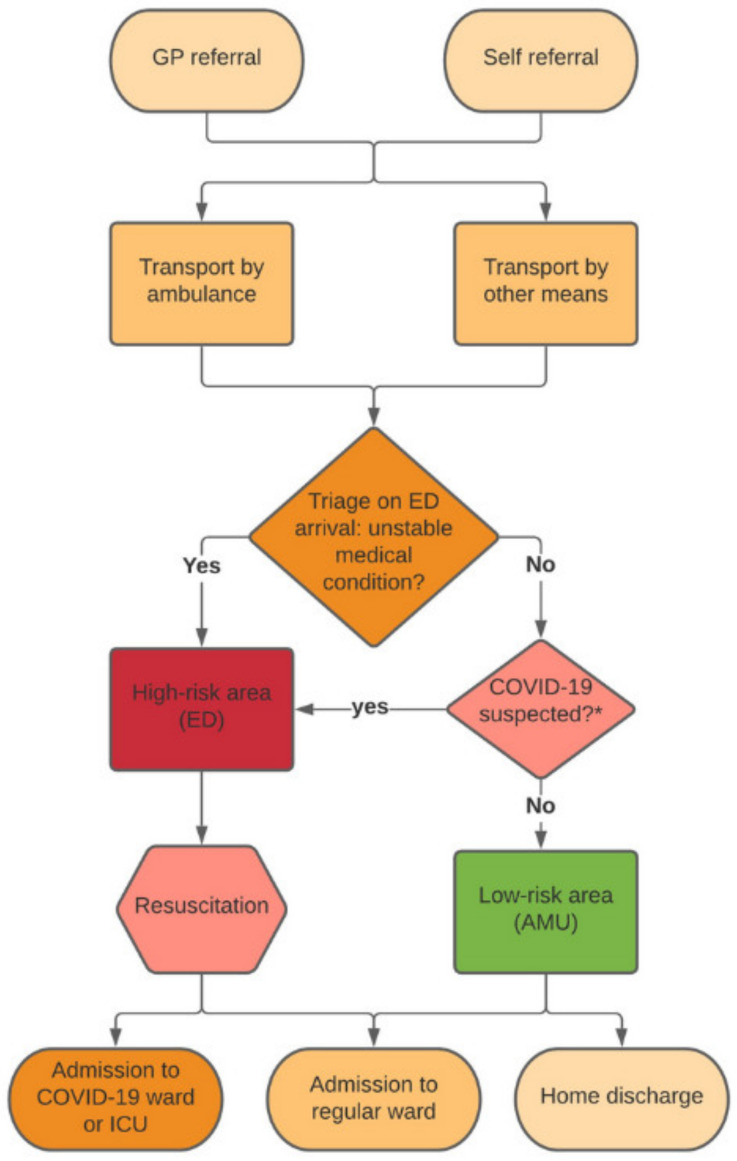
Flowchart of pre-triage screening COVID-19 suspected patients in the emergency department. GP, general practitioner; ED, emergency department; AMU, acute medical unit; ICU, intensive care unit. * Based on latest guidelines concerning symptoms, known COVID-19 contacts and travel history. Figure adapted from Barten et al. [25].

**Table 1 ijerph-18-09271-t001:** Baseline characteristics of patients allocated to AMU or ED.

	Total (*n* = 1287)	AMU (*n* = 525)	ED (*n* = 762)	*p*-Value
**General ^α^**				
Age (years)	67 (51–78)	60 (42–76)	69 (57–78)	*p* < 0.001
Male	709 (55.1%)	266 (50.7%)	443 (58.1%)	*p* = 0.008
**Symptoms ^β^**				
Respiratory tract Gastrointestinal tract Urogenital tract Neurological tract Cardiac tract Non-specific symptoms Trauma related symptoms Other symptoms	547 (42.5%) 184 (14.3%) 55 (4.3%) 122 (9.5%) 49 (3.8%) 112 (8.7%) 264 (20.5%) 131 (10.2%)	5 (1.0%) 111 (21.1%) 36 (6.9%) 75 (14.3%) 18 (3.4%) 18 (3.4%) 214 (40.8%) 78 (14.9%)	542 (71.1%) 73 (9.6%) 19 (2.5%) 47 (6.2%) 31(4.1%) 94 (12.3%) 50 (6.6%) 53 (7.0%)	*p* < 0.001 *p* < 0.001 *p* < 0.001 *p* < 0.001 *p* = 0.556 *p* < 0.001 *p* < 0.001 *p* < 0.001
**Primary specialty**				
Internal medicine Pulmonology Gastroenterology Surgery Orthopaedics Cardiology Neurology Urology Other	342 (26.6%) 318 (24.7%) 77 (6.0%) 286 (22.2%) 77 (6.0%) 13 (1.0%) 118 (9.2%) 44 (3.4%) 12 (0.9%)	78 (14.9%) 6 (1.1%) 51 (9.7%) 207 (39.4%) 71 (13.5%) 2 (0.4%) 74 (14.1%) 28 (5.3%) 8 (1.5%)	264 (34.6%) 312 (40.9%) 26 (3.4%) 79 (10.4%) 6 (0.8%) 11 (1.4%) 44 (5.8%) 16 (2.1%) 4 (0.5%)	*p* < 0.001 *p* < 0.001 *p* < 0.001 *p* < 0.001 *p* < 0.001 *p* = 0.061 *p* < 0.001 *p* = 0.002 *p* = 0.067
**Triage ^γ^**	***n* = 1272**	***n* = 512**	***n* = 760**	
Red/Orange Yellow Green/Blue	328 (25.5%) 570 (44.3%) 374 (29.0%)	59 (11.2%) 237 (45.1%) 216 (41.2%)	269 (35.3%) 333 (43.7%) 158 (20.7%)	*p* < 0.001 *p* = 0.609 *p* = 0.001
**COVID-19 RT-PCR**				
COVID-19 RT-PCR performed COVID-19 RT-PCR positive	501 (38.9%) 317 (24.6%)	28 (5.3%) 13 (46.4%)	473 (62.1%) 304 (64.3%)	*p* < 0.001 *p* = 0.002
**Clinical outcome**				
Admission rate	720 (55.9%)	191 (36.4%)	529 (69.4%)	*p* < 0.001
In-hospital mortality **^δ^**	107 (8.3%)	8 (1.5%)	99 (13.0%)	*p* < 0.001

^α^ Values are *n* (%) for ordinal variables and median (IQR) for continuous variables, unless otherwise specified. ^β^ Respiratory symptoms: dyspnoea (d’effort), coughing, painful respiration, sputum production, rhinorrhoea; Abdominal symptoms: stomach ache/cramps, nausea, vomiting, diarrhoea; Urological symptoms: oliguria, dysuria, pollakisuria, haematuria; Neurological symptoms: headache, sensibility loss, loss of motor function, dysarthria, vision problems; Cardiac symptoms: chest pain, palpitations, orthopnoea, fainting; Non-specific symptoms: fever, chills, loss of smell and taste, delirium, loss of appetite, weight loss, muscle ache, muscle weakness, anaemia; Other symptoms: symptoms not able to be classified in categories as above mentioned. ^γ^ According to Manchester Triage System (MTS). ^δ^ 30 Day in-hospital mortality.

**Table 2 ijerph-18-09271-t002:** Table (2 × 2) of ED and AMU patients with RT-PCR performed.

Allocation Area	COVID-19 Positive	COVID-19 Negative	Total
**ED ^α^**	304	169	473
**AMU ^β^**	13	15	28
**Total**	317	184	501
**Sensitivity**	95.9% (93.1–97.8%)		
**Specificity**	8.2% (4.6–13.1%)		
**PPV ^γ^**	64.3% (63.1–65.4%)		
**NPV ^δ^**	53.6% (36.0–70.3%)		
**Accuracy**	63.7% (59.3–67.9%)		

^α^ RT-PCR performed in 473/762 (62.1%) patients. ^β^ PCR performed in 28/525 (5.3%) patients. ^γ^ PPV = positive predictive value. ^δ^ NPV = negative predictive value.

**Table 3 ijerph-18-09271-t003:** Baseline characteristics of the 13 COVID-19 RT-PCR positive patients initially allocated to AMU.

	*n* = 13
**General ^α^**	
Age (years)	76 (65–92)
Male	7 (53.8%)
**Symptoms ^β^**	
Respiratory tract Gastrointestinal tract Urogenital tract Neurological tract Non-specific symptoms Trauma	2 (15.4%) 7 (53.8%) 1 (7.7%) 3 (23.1%) 6 (46.2%) 2 (15.4%)
**Primary specialty**	
Internal medicine Gastroenterology Pulmonology Neurology Surgery Orthopaedics Urology	4 (30.8%) 1 (7.7%) 1 (7.7%) 2 (15.4%) 2 (15.4%) 2 (15.4%) 1 (7.7%)
**Triage ^γ^**	*n* = 12
Red/Orange Yellow Green/Blue	2 (16.7%) 7 (58.3%) 3 (25.0%)
**Vital signs**	
Peripheral oximetry ≤95% (*n* = 12) Respiratory rate ≥20 per minute (*n* = 10) Pulse rate >100 bpm (*n* = 12) Temperature <36.0 °C or ≥38.0 °C (*n* = 12) Glasgow Coma Scale <15 (*n* = 13)	3 (25.0%) 4 (40.0%) 3 (25.0%) 3 (25.0%) 4 (30.8%)
**Clinical outcome**	*n* = 11
Admission rate	8 (72.7%)
In-hospital mortality **^δ^**	3 (27.3%)

^α^ Values are n (%) for ordinal variables and median (IQR) for continuous variables, unless otherwise specified. ^β^ Symptoms are similar as described in Table 1. ^γ^ According to Manchester Triage System. ^δ^ 30 Day in-hospital mortality.

## Data Availability

The datasets used and/or analysed during the current study are available from the corresponding author on reasonable request.

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
