# Peer review of "Sort and Sieve: Pre-Triage Screening of Patients with Suspected COVID-19 in the Emergency Department"

_ijerph, 2021, doi:10.3390/ijerph18179271_

Round 1

Reviewer 1 Report

Hensgens et al. reported a retrospective study to evaluate the performance of the pre-triage screening method for suspected COVID-19 patients in a hospital in the Netherlands. They further analyzed the characteristics of patients who were initially undetected, but later diagnosed with COVID-19. By analyzing a total of 1,287 patients, they showed that the pre-triage screening accuracy was about 64%. They also found that the patients aged 65 or above, presented with gastro-intestinal or non-specific symptoms. Therefore, they suggest that the emergency department (ED) screening protocols should include non-specific symptoms in older patients. 

This is a well written manuscript, the authors presented their data very well.

However, there are a few things that need to be improved.

  1. Introduction:  the authors referred to the Manchester Triage System (MTS) throughout the tables, however, did not explain about MTS. It would be very helpful if the authors introduce MTS in the introduction.
  2. There is one thing that the authors need to mention/discuss more. RT-PCR method can detect viral RNA in the upper respiratory tract for a mean of 17 days after symptom onset. However, the viral culture from PCR positive upper respiratory tract samples has been rarely positive beyond nine days of illness. Therefore, for those patients who came to the ED, they might already be past nine days of illness. Therefore, where and when taking the sample is also important.
  3. The authors looked at patients aged 18 or above, but data presented here were focused on elderly patients. Is that because the young patients showed asymptomatic? If the authors have the data, it could be included.

Reviewer 2 Report

Thanks for giving me the opportunity to review "Sort and sieve: pre-triage screening of patients with suspected COVID-19 in the emergency department". the paper is interesting and provides novel insights into the existing literature. There are some concerns that should be considered before publication.

the authors are needed to improve the introduction section by adding the latest and relevant literature. they also needed to elaborate on the findings of prior research they cited, for example, what has been done in those studies? what are their findings? etc. it would help them to rationale their study.

add the potential findings/contribution at the end of the introduction section.

the methodology section is discussed very well.

the authors link their results in a good manner to prior research in the discussion section.

Including the limitations of the study is highly appreciated. 

the conclusion is well-linked to the findings of the study.

Reviewer 3 Report

Interesting and well written article

Reviewer 4 Report

Congratulation

Good manuscript that proved that during a pandemic, scarce diagnostic tests should be used expediently and one might question the added value of testing asymptomatic or pre-symptomatic patients in an ED setting.

We should remember that proper balance between PPE and hospital beds volume were important in the beginning of 2020. The protocols were changed in next months and we know that some countries developed programs of massive testing. There is no ready perfect receipt for pandemic. It should be included in limitations too.

Please provide COVID-19 not covid-19 ex. line 44.

Round 2

Reviewer 2 Report

Thanks for giving me the opportunity to review the revised version of the manuscript.

The authors suffetiantly answered modified the introduction section.

I'm highly satisfied with their revision.

I accept the paper in its present form.